# Identification and Characterization of Novel Small-Molecule SMOX Inhibitors

**DOI:** 10.3390/medsci10030047

**Published:** 2022-08-30

**Authors:** Amelia B. Furbish, Ahmed S. Alford, Pieter Burger, Yuri K. Peterson, Tracy Murray-Stewart, Robert A. Casero, Patrick M. Woster

**Affiliations:** 1Department of Drug Discovery and Biomedical Sciences, College of Pharmacy, Medical University of South Carolina, 70 President St., Charleston, SC 29425, USA; 2Sidney Kimmel Comprehensive Cancer Center, Department of Oncology, Johns Hopkins School of Medicine, 1650 Orleans St. Room 551, Baltimore, MD 21287, USA

**Keywords:** spermine oxidase, polyamines, oxidative stress, acrolein, excitotoxicity, polyamine oxidase, neuronal injury

## Abstract

The major intracellular polyamines spermine and spermidine are abundant and ubiquitous compounds that are essential for cellular growth and development. Spermine catabolism is mediated by spermine oxidase (SMOX), a highly inducible flavin-dependent amine oxidase that is upregulated during excitotoxic, ischemic, and inflammatory states. In addition to the loss of radical scavenging capabilities associated with spermine depletion, the catabolism of spermine by SMOX results in the production of toxic byproducts, including H_2_O_2_ and acrolein, a highly toxic aldehyde with the ability to form adducts with DNA and inactivate vital cellular proteins. Despite extensive evidence implicating SMOX as a key enzyme contributing to secondary injury associated with multiple pathologic states, the lack of potent and selective inhibitors has significantly impeded the investigation of SMOX as a therapeutic target. In this study, we used a virtual and physical screening approach to identify and characterize a series of hit compounds with inhibitory activity against SMOX. We now report the discovery of potent and highly selective SMOX inhibitors **6** (IC_50_ 0.54 μM, Ki 1.60 μM) and **7** (IC_50_ 0.23 μM, K_i_ 0.46 μM), which are the most potent SMOX inhibitors reported to date. We hypothesize that these selective SMOX inhibitors will be useful as chemical probes to further elucidate the impact of polyamine catabolism on mechanisms of cellular injury.

## 1. Introduction

The dysregulation of polyamine metabolism has been implicated as a key mechanism of injury across multiple forms of clinically challenging pathologies, including acute [1,2] and chronic neuronal injury [3,4], renal failure [5], diabetes [6], and carcinogenesis [7,8] (Figure 1). Of the enzymes within the polyamine pathway, the catabolic enzyme spermine oxidase (SMOX) is of particular interest as it is subject to induction in response to infection [9], neuronal excitotoxicity [10,11], ischemia [12], and oxidative stress [13,14]. SMOX is found in the nucleus and cytoplasm of mammalian cells where it catalyzes the oxidation of spermine to spermidine, resulting in the production of H_2_O_2_ and 3-aminopropanal (3-AP). Under physiologic conditions, 3-AP undergoes spontaneous conversion to acrolein, a highly toxic aldehyde with the ability to form adducts with DNA, inactivate vital cellular proteins, and crosslink cellular structural components. In addition to the potential for secondary injury associated with this subsequent increase in toxic metabolic byproducts, the upregulation of SMOX results in the depletion of intracellular spermine, which introduces an additional mechanism of cellular insult. Spermine possesses a variety of protective properties, including the ability to act as a free radical scavenger [15,16] and induce a voltage-dependent blockade of AMPA and NMDA receptors [17], which may represent an underlying mechanism for the neuroprotective role of polyamines.

Spermine catabolism to spermidine may also occur through a two-step enzymatic process, in which spermine is first converted to *N*^1^-acetylspermine by spermidine/spermine *N*^1^-acetyltransferase (SSAT) and then oxidized to spermidine by *N*^1^-acetylpolyamine oxidase (PAOX), resulting in the production of H_2_O_2_ and 3-acetamidopropanal (3-AAP). However, unlike SMOX, PAOX is constitutively expressed and does not undergo upregulation in response to acute or chronic cellular injury, and SMOX, not PAOX, has been implicated as the primary source of cytotoxic H_2_O_2_ production in cells treated with polyamine analogs [18]. In addition, spermine catabolism by SMOX has been identified as a significant endogenous source of acrolein, along with non-specific lipid peroxidation [4,19,20]. Acrolein is the most toxic endogenously produced aldehyde and the most toxic byproduct of polyamine metabolism [4]. Of note, 3-AAP produced by PAOX does not produce cytotoxicity, even at mM concentrations [4]. 

While there is significant evidence implicating SMOX as a therapeutic target, currently available SMOX inhibitors lack potency, selectivity, and acceptable pharmacokinetic parameters, limiting their use as both in vivo probe compounds and potential therapeutics (Figure 2). The most commonly described and well-characterized SMOX inhibitor, MDL 72527 (*N*^1^,*N*^4^-bis(2,3-butadienyl)-1,4-butanediamine, Ki = 63 µM [21], IC_50_ = 89–100 µM [22,23]), acts as an irreversible inhibitor of SMOX. MDL72527, as well as other reported SMOX inhibitors such as SI-4650 (Ki = 382 µM, IC_50_ = 289 µM) [24] and 2,11-Met_2_Spm (IC_50_ = 169 µM) [22], lack the sufficient potency for translation to the clinic. Inhibitors such as methoctramine (Ki = 1.2 µM) exhibit improved potency but are selective for PAOX over SMOX and display off-target activity at muscarinic M_2_ receptors [25,26]. In addition to the inhibitors described above, a variety of compounds containing guanidine or guanidine-like moieties have been shown to act as potent SMOX inhibitors with varying degrees of selectivity (Figure 2). The guanidine-based alkyl amines *N*-prenylagmatine (Ki = 0.46) and guazatine (iminoctadine) (Ki = 0.4) [21] (Figure 2) are limited by both a lack of selectivity and a lack of structurally modifiable moieties necessary for the targeted improvement of drug-like characteristics. Similarly, the bis-guanidine containing SMOX inhibitor, chlorhexidine (Ki = 0.55 µM) [27], and the 1,2,4-diaminotriazole containing SLH-59 (IC_50_ = 25.7 µM) [23] are limited by a lack of selectivity for SMOX, and poor pharmacologic properties such as toxicity, which is associated with systemic use (chlorhexidine) and poor solubility (SLH-59). In light of these observations, the goal of this study was the identification of a novel structural scaffold on which to build potent and selective SMOX inhibitors. Importantly, such inhibitors would be of value as chemical tools for validating SMOX as a therapeutic target and for understanding the key pathogenic mechanisms associated with the dysregulation of polyamine metabolism. Although a crystal structure for SMOX alone or bound to a suitable inhibitor is not yet available, there is some information available concerning the active site [25,28,29,30]. Molecular modeling, site-directed mutagenesis, and biochemical characterization studies of the SMOX enzyme–substrate complex have identified Glu216–Ser218 as putative active site residues responsible for SMOX substrate specificity [30]. Due to the lack of SMOX structural information, we developed a SMOX homology model based on these data for use in the screening and structure-based design of novel SMOX inhibitors. Hit compounds were identified by in silico screening methods and confirmed for activity via enzymatic assay. Additional inhibitors were identified through structural similarity searching and characterized for potency and selectivity against related FAD-dependent amine oxidase enzymes lysine-specific demethylase 1 (LSD1) and monoamine oxidase enzymes A and B (MAO-A and MAO-B). Docking simulations and computational methods were used to describe drug-like characteristics and predict the mechanism of binding. Select inhibitors were then assessed for cellular toxicity and the ability to alter cellular response. 

## 2. Materials and Methods

### 2.1. Molecular Docking, In Silico Screen, and Similarity Searching

A PSI_BLAST of the human SMOX sequence (Uniprot: Q9NWM0) was performed against Zea mays polyamine oxidase (PAO) (PDB 1B37, 1.9 Å), which was identified as the best-suited template structure (E-value: 2-21) for the development of a SMOX homology model. This result is in agreement with previous work by Tavladoraki [29]. Protein alignment was generated between the human SMOX and the *Z. mays* PAO using the program Muscle [31], which indicated that the two proteins possess 24% sequence identity and 45% sequence similarity. Homology models were then built using Modeller 9v14 [32] using the *Z. mays* PAO structure (PDB 1B37) [33] as a template. Amino acids were numbered with the initiating methionine set to 1. The models were subjected to quality analysis using the PDBsum generator (http://www.ebi.ac.uk/pdbsum, accessed on 22 July 2022) [34]. The best model (Figure 3) demonstrated a G-Factor value of −0.32 (values below −0.5 are considered unusual). Docking simulations to predict binding were performed and visualized using molecular modeling software (MOE and PyMol). ADMET properties were calculated using a variety of software resources including SwissADME, CDD Vault, and Chemaxon. Following the identification of hit molecules, a similarity search of the South Carolina Compound Collection (SC^3^)—a fully annotated chemical library consisting of 130,000 proprietary compounds—was conducted using CDD Vault (https://www.collaborativedrug.com/, accessed on 22 July 2022) with a cutoff value at a Tanimoto coefficient of ≥0.7.

### 2.2. Determination of SMOX Enzyme Activity

As the catabolism of spermine by SMOX results in the stoichiometric production of H_2_O_2_, enzyme activity was quantified by chemiluminescence generated by the horseradish peroxidase-catalyzed oxidation of luminol in 96-well plates (Costar #3912; white, flat bottom), as previously described [35]. All assays were performed in 0.083 M glycine buffer at a pH of 8 using 0.3 µg/mL of purified SMOX enzyme and 2 mM of spermine substrate. Phosphate buffer was not used to prevent the formation of polyamine–phosphate complexes [36]. Enzyme reactions were initiated by the addition of SMOX and HRP to all other assay components; inhibitors were not pre-incubated with enzyme. Triplicate determinations were obtained for each measurement by recording the luminescence (AUC) over 60 s. Hit compounds were then confirmed by monitoring the assay over 600 s. Each triplicate assay point contained a 4th control well to monitor and normalize the data with regards to luminescent decay over time. The compounds were initially screened for activity at a concentration of 20 µM. Enzyme activity for each compound was compared to a corresponding blank (measured in the absence of substrate), and % inhibition was determined in comparison to vehicle control (defined as 100% activity). The known SMOX inhibitor, MDL-72527, was used as a control. The hit compounds were then characterized by IC50 and kinetic analyses. Percent inhibition values are reported as triplicate mean % SMOX inhibition ± SD compared to the vehicle control. IC50 values were calculated in GraphPad Prism 9.3 software (Graph-Pad, San Diego, California) using a non-linear regression analysis ([Inhibitor] vs. normalized response—variable slope). Data for all enzyme assays were generated using a SpectraMax M5 plate reader (Molecular Devices) equipped with SOFTmax PRO 7.0.3 software. 

### 2.3. Mechanism of Inhibition and Enzyme Kinetics

The mechanism of enzyme inhibition was determined using the HRP-luminol coupled enzyme assay described above. Reaction velocity (max RLU/sec over 10 min) was measured across multiple substrate and inhibitor concentrations, and individual triplicate values were plotted using a Lineweaver–Burk transformation (double-reciprocal) in GraphPad Prism 9.3 software (Graph-Pad, San Diego, CA, USA). A global fit analysis was performed for competitive enzyme inhibition, which was used to calculate Ki (R^2^ ≥ 0.98). 

### 2.4. Determination of Enzyme Selectivity

To determine selectivity, the compounds were tested for activity against the related FAD-dependent amine oxidase enzymes LSD1, MAO-A, and MAO-B. LSD1 activity was assessed using a commercially available assay kit (#700120, Cayman Chemical, Ann Arbor, MI, USA), in which the enzyme activity was determined by the horseradish peroxidase-catalyzed oxidation of 10-acetyl-3,7-dihydroxyphenoxazine to the highly fluorescent resorufin in response to H_2_O_2_ production. Monoamine oxidase activity was determined using recombinant human MAO-A and MAO-B (M7316-1VL & M7441-1VL, Sigma-Aldrich, St. Louis, MO, USA) and a commercially available assay kit (#V1401, MAO-Glo, Promega, Madison, WI, USA), which utilizes the MAO-catalyzed oxidation of a luciferin derivative coupled with esterase activity to produce luminescence. Fluorescent and luminescent readout was obtained using a SpectraMax M5 plate reader (Molecular Devices) equipped with SOFTmax PRO 7.0.3 software. All the compounds were tested at a concentration of 20 µM and compared to tranylcypromine (TCP) and the vehicle control. Values were reported as mean % inhibition ± SD of triplicate assay wells normalized to background activity and compared to the vehicle control. 

### 2.5. Cell Culture 

SH-SY5Y neuroblastoma cells purchased from ATCC (Manassas, VA, USA) were cultured in Dulbecco’s modified Eagle medium (DMEM) without phenol red and supplemented with 10% heat inactivated human AB serum, 4 mM L-glutamine, and 1% penicillin/streptomycin. As the oxidation of spermine by various polyamine oxidases has been documented in cell culture containing ruminant serum, such as in fetal bovine serum (FBS) and horse serum, human AB serum was used to prevent assay interference [37]. The cells were maintained in 5% CO_2_ humidified atmosphere at 37 °C. Cell media was replaced every 2 days and the cells were discarded after a maximum of 15 passages. 

### 2.6. Cytotoxicity Analysis

The SH-SY5Y neuroblastoma cells were plated in 96-well plates at a density of 30,000 cells/well and allowed to reach 80% confluence, followed by treatment with varying concentrations of inhibitor ranging from 0 to 10 mM for 12 h. Cell viability was assessed by standard MTS assay, in which MTS reagent was added to each well at a concentration of 0.33 mg/mL and incubated at 37 °C for 3 h. Viability was monitored and confirmed by a visual inspection of the cells. Absorbance was read at 490 nm and the absorbance values were normalized to background wells containing varying concentrations of inhibitor, cell media, and MTS reagent. Colorimetric readout was obtained using a SpectraMax M5 plate reader (Molecular Devices) equipped with SOFTmax PRO 7.0.3 software. 

### 2.7. Measurement of Cell Viability Following Excitotoxic Stress

The SH-SY5Y neuroblastoma cells were plated at a density of 1 × 10^6^ cells/mL in 96-well plates (Costar #9018; clear flat bottom) and allowed to adhere overnight. Media was aspirated and the cells were then treated with low-serum culture media (containing 5% human AB serum), 80 mM of glutamate, and varying concentrations of compound. The glutamate and test compounds were added simultaneously with no pre-incubation period. Cell viability was measured at 16 h and 24 h post-treatment by a standard MTS assay (0.33 mg/mL incubated at 37 °C for 3 h). Absorbance was read at 490 nm following 3-h incubation with MTS reagent. Cell viability was compared to wells without added glutamate (100% viability) and wells with glutamate and no experimental treatment compound. The known SMOX inhibitor, MDL 72527, was used as a control. The statistical analysis was conducted in GraphPad Prism 9.3 software (Graph-Pad, San Diego, CA, USA) using a one-way ANOVA with multiple comparisons (α = 0.05). 

### 2.8. Reactive Oxygen Species (ROS) Production in SH-SY5Y Cells 

To measure intracellular ROS production following acute excitotoxic stress, a CM-H2DCFDA (Invitrogen) probe was used. The SH-SY5Y cells were incubated in the presence of dichlorodihydrofluorescein diacetate (CM-H2DCFDA, Invitrogen), 20 µM in HBSS, at 37 °C for 30 min. The cells were then pelleted and washed with HBSS buffer (without phenol red, magnesium, or calcium). Phosphate buffer was not used to prevent the formation of polyamine–phosphate complexes [36]. The cells were then resuspended in 96-well plates (Greiner #655086; black walled, cleat flat bottom) at a concentration of 1 × 10^6^ cells/mL in HBSS containing 0–100 µM of inhibitor and 8 mM of glutamate. Dichlorodihydrofluorescein (DCF) fluorescence was monitored over 1 h (ex 488 nm, em 530 nm). The increase in fluorescence was reported as the % activity of control wells in the absence of the inhibitor. Fluorescent readout was obtained using a SpectraMax M5 plate reader (Molecular Devices) equipped with SOFTmax PRO 7.0.3 software. All test compounds were assessed for fluorescent activity at 530 nm prior to experimentation. The statistical analysis was conducted in GraphPad Prism 9.3 software (Graph-Pad, San Diego, California) using a one-way ANOVA with multiple comparisons (α = 0.05).

## 3. Results

### 3.1. In Silico and Enzymatic Screening Process and Structural Similarity Searching

Using the SMOX homology model described above, a virtual docking screen of 130,000 sample compounds from our in-house compound library, the SC^3^, was performed. Of the 130,000 compounds that underwent in silico screening, the 20 compounds with the lowest predicted binding energy (−21.91 to −19.06 kcal) underwent enzymatic screening at 20 µM. The hit compounds (defined as achieving ≥ 90% inhibition at 20 µM) were further screened for potency and reproducibility, resulting in the identification of two hit compounds adhering to an overall similar general structure (Figure 4A). In an effort to identify additional inhibitors and further elucidate the influence of structural variation on inhibitory activity, initial hit structures were used to perform a structural similarity search within the SC^3^ (Tanimoto coefficient ≥ 0.7), resulting in the identification of a total of seven structurally related compounds (compounds **1**–**7**, Table 1). All the hit compounds adhered to a common structural scaffold comprised of a β-hydroxylated phenethylamine core with an alkyl chain linker and terminal guanidine moiety. All seven compounds also maintained inhibitory activity against SMOX despite minor structural variations in the placement of aromatic substituents and alkyl chain length (Figure 4B).

### 3.2. Enzyme Inhibition Studies

Following initial screening, the hit compounds were confirmed for inhibitory activity by monitoring SMOX-dependent ROS production over 10 min (Figure 5A,B). All guanidine-based hit compounds (**1**–**7**) exhibited a significant and rapid inhibition of H_2_O_2_ production (as measured by HRP catalyzed oxidation of luminol) compared to MDL-72527 and vehicle controls. The compounds were not pre-incubated with enzyme, and luminescence reads were taken at 15 s intervals immediately following the addition of substrate (spermine). The rapid onset of inhibition observed in the presence of all test compounds without a pre-incubation period suggests a lack of time-dependent inhibition, in comparison with MDL 72527. In addition, given that the concentration of substrate used in this assay was two orders of magnitude larger than the concentration of inhibitor (i.e., 2 mM of spermine vs 20 µM of inhibitor), these data suggest a higher affinity for the inhibitor than the substrate. This is supported by the kinetic studies of compounds **6** and **7**, which indicate a competitive mechanism of inhibition with Ki values of 1.6 µM and 0.46 µM, respectively (Figure 5C,D). This is in agreement with previous studies of other guanidine-based SMOX inhibitors, including chlorhexidine, *N*-prenylagmatine, and guazatine, which behave as competitive inhibitors of SMOX [21,27].

Compounds **1**–**7** all demonstrated inhibitory activity against SMOX at 20 µM with % inhibition values ranging from 88% to 100% and IC_50_ values in the nM range (0.232–10.59 μM; Figure 6A). Six of the seven compounds are benzenediol substituted compounds with two hydroxyl substituents on either the 2,5-, 3,5-, or 3,4- positions on the benzene ring. By contrast, compound **6** consists of a para-hydroxyl substituent surrounded by two methyl groups in the 3 and 5 position on the benzene ring. Compounds containing either a hydroquinone or catechol moiety appeared to be slightly more potent than compounds containing a resorcinol moiety. 

The most potent compounds assessed were compounds **4** and **7**; however, when tested for inhibitory activity against related FAD dependent amine oxidase enzymes (LSD1, MAO-A, and MAO-B), both compounds, particularly compound **7,** appeared to exhibit off-target activity at all three enzymes, with just over 73% inhibition for LSD1, 45% inhibition for MAO-A, and 29% inhibition for MAO-B. Although all compounds possessed the highest affinity for SMOX, similar trends in potency, i.e., increased alkyl chain length and the presence of a hydroquinone or catechol moiety, were observed for LSD1. In addition to compound **7**, compounds **4** and **5** appeared to have some inhibitory activity towards MAO-A and MAO-B, which was not surprising given the preponderance of catechol-based endogenous substrates of these enzymes (Figure 6B).

### 3.3. Molecular Docking and Computational Analysis

Molecular docking simulations were conducted using MOE and visualized using PyMOL (see Materials and Methods). Figure 7 shows compounds **6** and **7** in the lowest energy docking conformation, which were determined using the SMOX homology model described above. Compounds **6** and **7** was chosen due to their high potency against SMOX, which suggested they were the most stable active site ligands. Docking simulations indicated significant polar contact interactions for **6** with PHE222, GLU208 and HIS82 (Figure 7A), and for **7** with GLU208. (Figure 7B). Graphical maps of these interactions appear in Figure 7C,D. The in silico analysis of compounds **1**–**6** (not shown) indicated that these compounds were bound in similar structural conformations, which further supports the kinetic analysis indicating a competitive mechanism of inhibition. 

### 3.4. Effect of Inhibitors on Cellular Response

Compounds **4**, **6** and **7** were selected for further analysis in cell-based experiments due to their demonstrated potency (**4** and **7**) and selectivity (**6**). While compound **6** was the least potent of the seven analogues by IC_50_, it possessed superior selectivity and toxicity profiles (Figure 8) when compared to compounds **4** and **7**. For this experiment, compounds **4**, **6** and **7** (IC_50_ values 0.23, 0.54 and 0.23 µM, respectively) were dosed at levels that were 40- to 40,000-fold higher than their enzymatic IC_50_ values. At these high dose levels, compounds **4** and **6** did not produce significant toxicity after 12-h exposure at doses below 1 mM. Of note, compound **7** exhibited significant toxicity at micromolar concentrations (LC_50_ = 0.13 μM), precluding its use for in vitro analysis. Compound **6** was ultimately selected for use in further studies given its excellent toxicity profile (LC_50_ = 5.8 mM) and lack of off-target effects. 

The dysregulation of polyamine metabolism, particularly through the upregulation of SMOX and production of the associated toxic metabolites, has been demonstrated to be a major contributing factor associated with excitotoxic injury [10,11,14,38,39,40,41,42,43,44]. In addition, several studies have demonstrated the neuroprotective effects of MDL 72527 in both in vitro and in vivo models of excitotoxic damage [35,36,37]. While glutamate has been shown to exert cytotoxic effects due to both oxidative damage and the NMDA-receptor-elicited dysregulation of calcium homeostasis in primary neurons, previous studies have indicated that glutamate exerts toxicity primarily through oxidative damage in SH-SY5Y neuroblastoma cell lines [45]. 

As this directly aligns with the effects of SMOX upregulation, the compounds were assessed for the ability to inhibit cytotoxicity in SH-SY5Y neuroblastoma cells exposed to glutamate (Figure 9). Cell viability was assessed by an MTS assay in SH-SY5Y neuroblastoma cells following 16- and 24-h incubation with 80 mM of glutamate and compound **6**. A glutamate concentration of 80 mM was chosen because this is the published Lzc50 for glutamate in SH-SY5Y cells [45,46,47]. Glutamate treatment significantly decreased cell viability at both 16 and 24 h in comparison to untreated control wells, which were designated as 100% viability (*p* < 0.0001). While all treatment groups resulted in higher measured viability than vehicle control at 16 h, particularly treatment with 100 µM of compound **6** (26.5% vs. 48.3% viability), statistical significance was not achieved. At 24 h incubation, treatment with compound **6** at a concentration of 1 mM significantly increased cell viability in comparison to both vehicle control (*p* = 0.0001) and MDL 72527 (*p* = 0.0003).

In addition to improved viability at 24 h, compound **6** reduced the production of ROS in SH-SY5Y cells following acute excitotoxic stress (8.0 mM of glutamate exposure) in a dose-dependent manner, as measured by CM-H2DCFDA at 1 h post treatment (Figure 10). Treatment with both 10 µM and 50 µM of compound **6** significantly decreased the production of ROS in comparison to the vehicle control (*p* < 0.0001). In addition, a dose-dependent effect was seen, with the higher dose (50 µM) achieving a significant reduction in ROS over the 10 µM dose (*p* = 0.0004). 

## 4. Discussion

The goal of this study was to identify and characterize potent and selective SMOX inhibitors suitable for use as chemical probes to further elucidate the mechanisms of cellular injury associated with SMOX induction and overall polyamine catabolism. Through virtual and physical screening methods, we were able to identify a series of seven SMOX inhibitors, some of which represent potent and highly selective SMOX inhibitors with IC_50_ values as low as 232 nM (Table 1). All the hit compounds adhere to a common structural scaffold comprised of a β-hydroxylated phenethylamine core with an alkyl chain linker and terminal guanidine moiety and include structural variations in the placement of aromatic substituents and alkyl chain length. All compounds maintained inhibitory activity for SMOX, despite minor structural variations, indicating the discovery of a fundamentally novel pharmacophore with the ability to tolerate further chemical modification in the development of targeted therapies suitable for in vivo studies, and ultimately, to aid in the development of compounds suitable for clinical evaluation. 

Although compounds **4** and **7** demonstrated the lowest IC_50_ values, their utility is somewhat limited by off-target activity for the related FAD-dependent amine oxidase enzymes LSD1, MAO-A, and MAO-B. In addition, the presence of the catechol (**4**) and hydroquinone (**7**) moieties present on these compounds pose a potential risk of undesirable pharmacokinetic parameters, such as significant metabolism and rapid elimination seen with catechol-based drugs, and potential for the metabolic oxidation of hydroquinone moieties to toxic *p*-benzoquinones, as evidenced by the cytotoxic effects seen at µM concentrations of compound **7** in cell-based assays. Alternatively, compound **6** maintained both potency and selectivity for SMOX without the associated cytotoxic effects. In addition, compound **6** was able to rescue cell viability in the setting of excitotoxic stress and dose dependently decrease oxidative stress in cell-based studies, suggesting potential utility as an in vivo probe compound for use in further studies. 

In addition to the use as probe compounds for the further investigation of SMOX as a therapeutic target, the characterization of this set of guanidine-based inhibitors compliments previous work documenting the activity of structurally related inhibitors, in order to further elucidate the structure activity relationship (SAR) characteristic of potency and selectivity for SMOX, particularly given the prominence of guanidine-based compounds previously reported throughout the literature. This is of particular utility since the crystal structure of SMOX has not yet been solved, and therefore, little is known regarding the exact mechanism of binding to the enzyme. 

While the series of compounds described above represents a significant advancement in the discovery of potent and selective inhibitors for use as probe compounds in future studies, additional derivatization and analogue testing is required to elucidate optimal structural requirements indicative of potency and selectivity for the given target enzyme. In addition, pharmacokinetic factors affecting translational feasibility must first be addressed in the design and synthesis of future derivatives aimed at in vivo use, particularly in the setting of compounds intended for use as neuroprotective agents. 

## Figures and Tables

**Figure 1 medsci-10-00047-f001:**
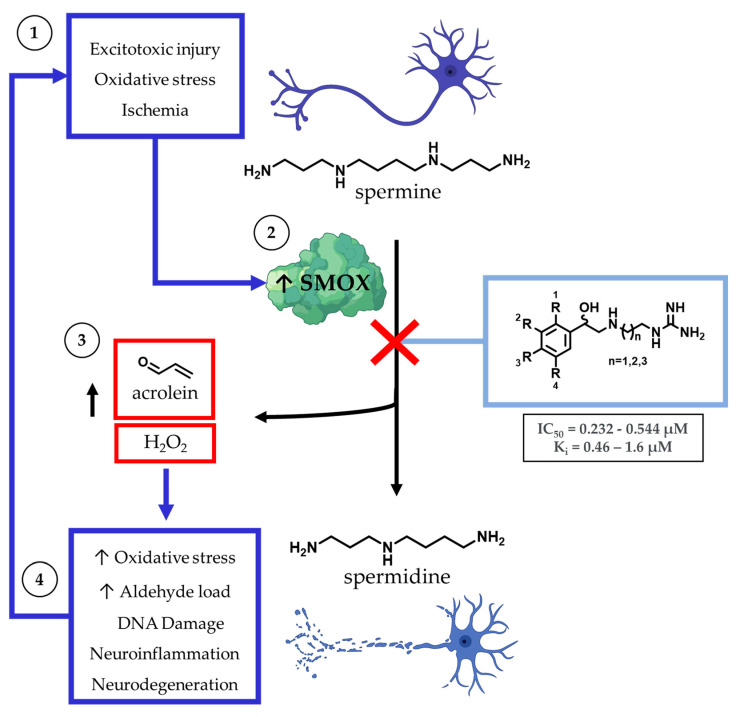
Within the central nervous system, the major intracellular polyamines spermine and spermidine are stored in astrocytes and, when released, have the potential to act as free radical scavengers in response to neuronal damage. Spermine catabolism is mediated primarily by spermine oxidase (SMOX), which becomes upregulated in a variety of cell types during ischemic, excitotoxic, and inflammatory states. Catabolism of spermine by SMOX results in the production of toxic byproducts, including H_2_O_2_ and acrolein.

**Figure 2 medsci-10-00047-f002:**
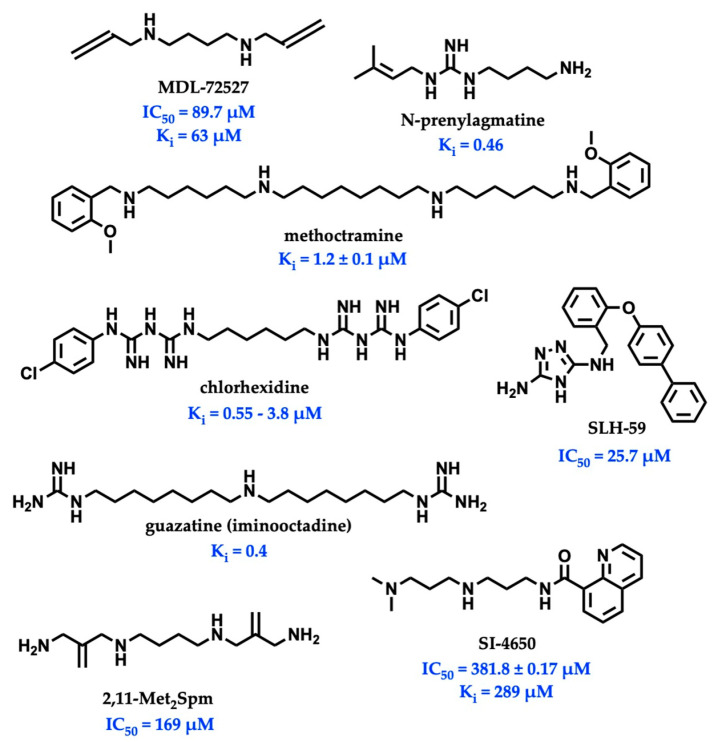
Structures and inhibitory values (IC_50_ and K_i_) of SMOX inhibitor reported in the literature.

**Figure 3 medsci-10-00047-f003:**
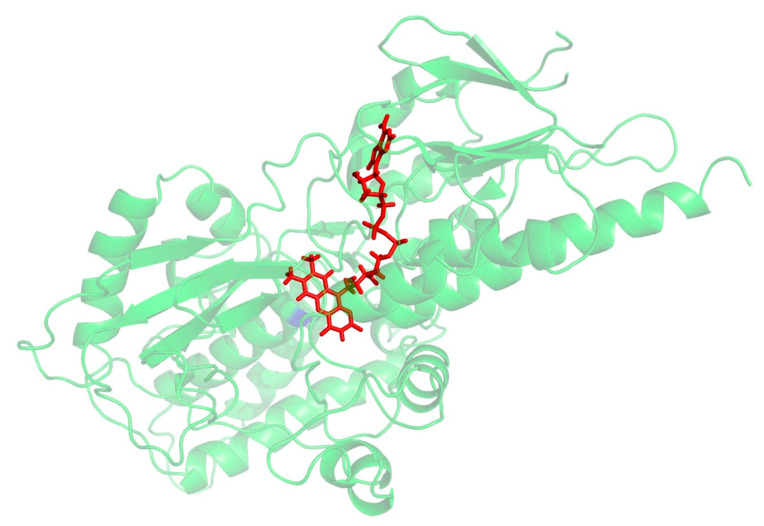
Preliminary homology model for SMOX. FAD is shown in red.

**Figure 4 medsci-10-00047-f004:**
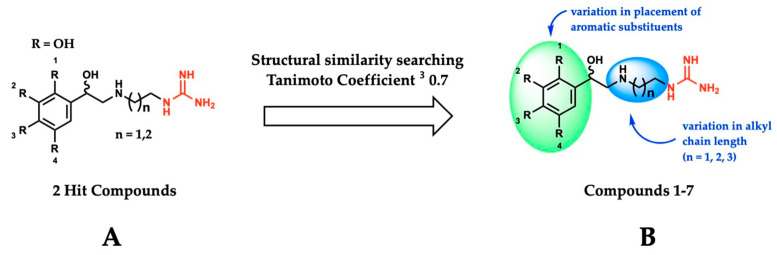
(**A**) General structure of hit compounds selected for further analysis from initial enzymatic screen and used for structural similarity searching. (**B**) Structural variation of guanidine-based hit compounds identified through initial enzymatic screening and structural similarity searching, resulting in compounds **1**–**7**.

**Figure 5 medsci-10-00047-f005:**
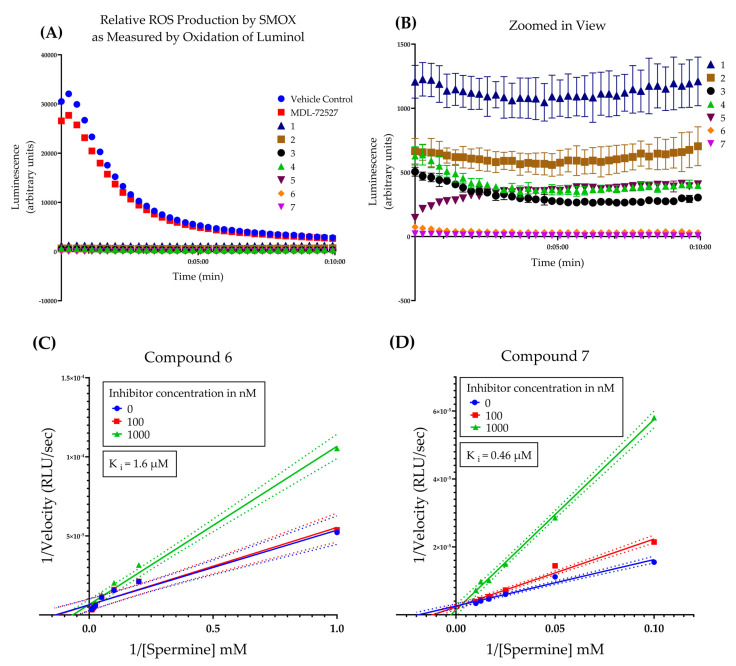
(**A**) Time course of the oxidation of spermine by SMOX as measured by hydrogen peroxide production coupled to HRP catalyzed oxidation of luminol over 10 min. All compounds (**1**–**7**) were assayed in triplicate at a concentration of 20 μM. Compounds were not pre-incubated with enzyme prior to reading. Luminescence was compared to vehicle control (shown in red) and MDL-72527 (shown in blue). Assay conditions: 0.083 M glycine buffer, pH = 8, 0.3 µg/mL of purified SMOX enzyme, 2 mM of spermine substrate, 25 °C. (**B**) Zoomed in view of data from panel A for compounds **1**–**7**. (**C**) Lineweaver–Burk plot of compound **6** across three doses (0, 0.1, and 1 µM) and varying concentrations of substrate [spermine] Ki = 1.6 µM. All R^2^ values for kinetic analysis were >0.98. (**D**) Lineweaver–Burk plot of compound **7** across three doses (0, 0.1, and 1 µM) and varying concentrations of substrate [spermine] K_i_ = 0.46 µM. All R^2^ values for kinetic analysis were > 0.98.

**Figure 6 medsci-10-00047-f006:**
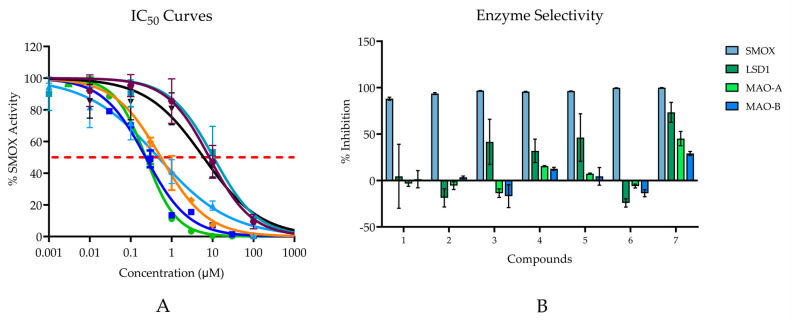
(**A**) IC_50_ values for hit compounds. Points are reported as mean % activity ± SD of triplicate values (AUC integrated over 60 s and normalized to vehicle control). Continuous line represents non-linear regression curve calculated in GraphPad Prism ([Inhibitor] vs. normalized response, variable slope, and least squares fit. (Compound **1** IC_50_ = 10.59 µM; Compound **2** IC_50_ = 8.30 µM; Compound **3** IC_50_ = 6.02 µM; Compound **4** IC_50_ = 0.23 µM; Compound **5** IC_50_ = 0.47 µM; Compound **6** IC_50_ = 0.54 μM; Compound **7** IC_50_ = 0.23 µM). For all reported values, R^2^ ≥ 0.95. (**B**) Enzyme selectivity for related FAD dependent amine oxidase enzymes at 20 µM.

**Figure 7 medsci-10-00047-f007:**
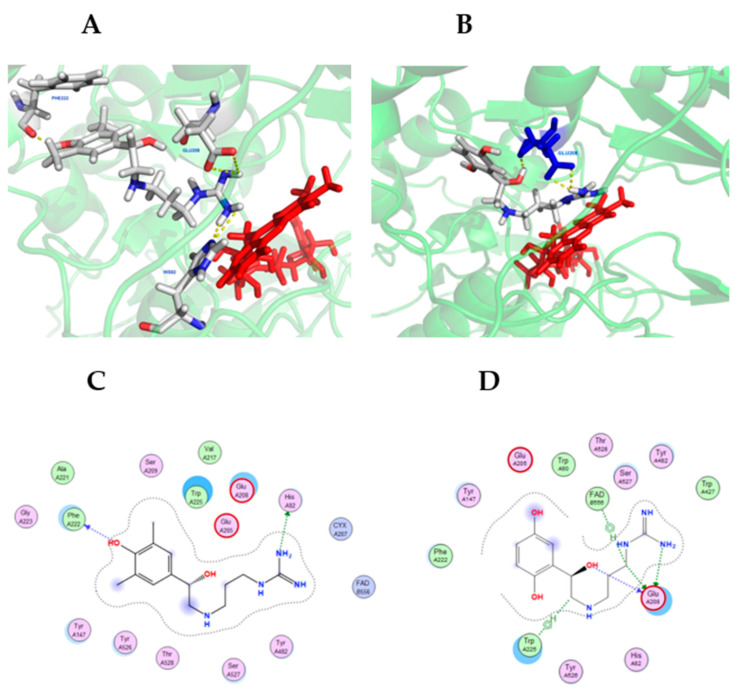
Hit compounds **6** (**A**) and **7** (**B**) in the binding-pocket of a SMOX homology model used for in silico screening. FAD is represented in red. The proposed docking conformation displays polar contact interactions (yellow) for **6** with PHE222, GLU208 and HIS82, and for **7** with GLU208. Panels C and D are graphical representations of SMOX homology model amino acid interactions for hit compounds **6** (**C**) and **7** (**D**) in their lowest energy docking pose.

**Figure 8 medsci-10-00047-f008:**
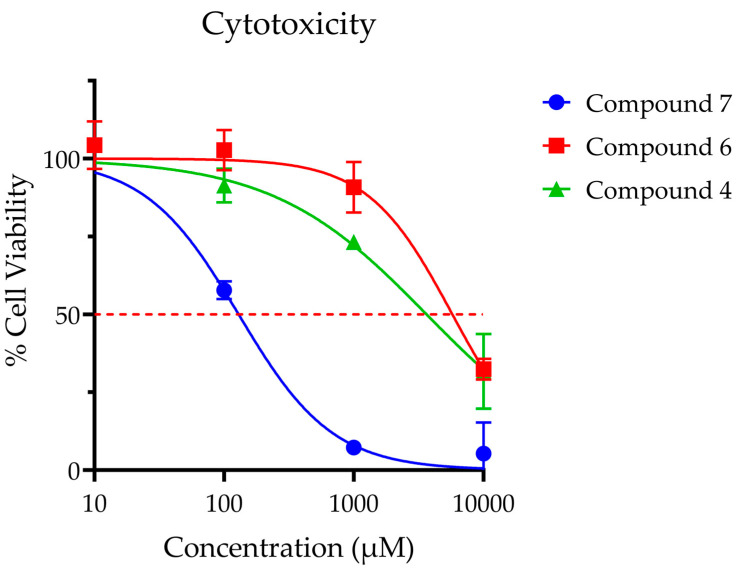
Cellular toxicity was assessed for SH-SY5Y neuroblastoma cells using standard MTS assay at 12 h post treatment. Points are reported at average % viability ± SEM normalized to vehicle control of 7 replicates. Continuous line represents non-linear regression curve calculated in GraphPad Prism ([Inhibitor] vs. normalized response, variable slope, and least squares fit. Calculated dose corresponding to 50% viability as follows: Compound **4** LC_50_ = 3.6 mM; Compound **6** LC_50_ = 5.8 mM; Compound **7** LC_50_ = 0.13 mM; Compound **6** LC_50_ = 5.8.

**Figure 9 medsci-10-00047-f009:**
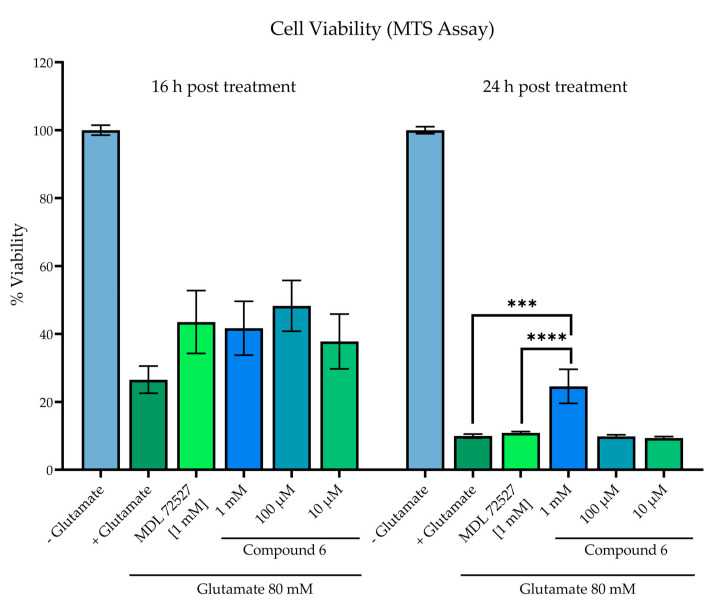
Cell viability in SH-SY5Y cells assessed by MTS assay in SH-SY5Y neuroblastoma cells following 16- and 24-h incubation +/− 80 mM of glutamate and test compound. Reported values represent the average of 12 replicate wells ± SEM. All groups receiving glutamate treatment were significant when compared to control wells without glutamate treatment, designated as 100% viability (*p* < 0.0001). **** *p* = 0.0001 vs. vehicle control; *** *p* = 0.0003 vs. MDL 72527.

**Figure 10 medsci-10-00047-f010:**
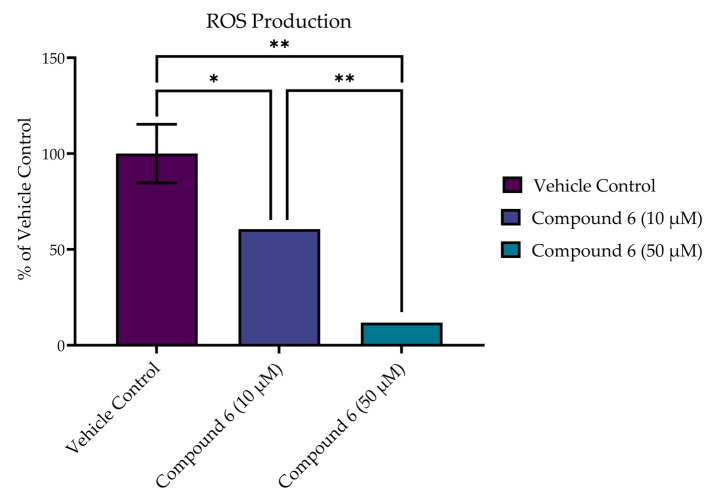
Compound **6** reduced production of ROS in SH-SY5Y cells following acute excitotoxic stress (8 mM of glutamate exposure) in a dose-dependent manner, as measured by CM-H2DCFDA. Reported values represent the average of 3 or 4 replicates ± SEM. * *p* = 0.0004; ** *p* < 0.0001.

**Table 1 medsci-10-00047-t001:** Inhibition of SMOX and related flavin-dependent oxidase enzymes LSD1, MAO-A, and MAO-B at 20 µM by compounds **1**–**7**. Tranylcypromine (TCP) and the known SMOX inhibitor MDL 72527 were used as controls. Values are reported as the average % inhibition of triplicate measurements ± SD. IC_50_ values of select inhibitors are also reported.

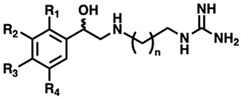									
						% Inhibition at 20 μM ± SD(IC_50_ in μM)			
Cmpd	R_1_	R_2_	R_3_	R_4_	n	SMOX	LSD1	MAO-A	MAO-B
**1**	H	OH	H	OH	1	88.1 ± 1.6 (10.59)	4.5 ± 34.5	−3.3 ± 3.0	1.4 ± 9.3
**2**	H	OH	H	OH	2	93.6 ± 1.0 (8.30)	−18.7 ± 9.8	−5.5 ± 4.2	3.5 ± 1.3
**3**	H	OH	H	OH	3	96.7 ± 0.2 (6.02)	41.6 ± 24.4	−13.7 ± 4.6	−16.9 ± 12.3
**4**	H	OH	OH	H	1	95.7 ± 0.3 (0.23)	31.9 ± 12.6	15.5 ± 0.5	12.6 ± 1.6
**5**	H	OH	OH	H	2	96.3 ± 0.1 (0.47)	46.2 ± 25.7	7.3 ± 0.7	4.5 ± 9.5
**6**	H	Me	OH	Me	2	99.7 ± 0.1 (0.54)	−24.2 ± 4.5	−6.0 ± 2.1	−13.7 ± 3.8
**7**	OH	H	H	OH	2	99.9 ± 0.1 (0.23)	73.4 ± 10.7	45.1 ± 7.7	29.1 ± 2.1
MDL-72527	-	-	-	-	-	10.2 ± 0.9	ND	ND	ND
TCP	-	-	-	-	-	ND	35.7 ± 14.7	92.6 ± 0.2	88.2 ± 7.9

## Data Availability

Data supporting reported results are contained within this article.

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
