# Peer review of "Identification and Characterization of Novel Small-Molecule SMOX Inhibitors"

_medsci, 2022, doi:10.3390/medsci10030047_

Round 1

Reviewer 1 Report

The manuscript entitled, “Identification and characterization of novel small-molecule SMOX inhibitors” by Woster et al is well written and describes the author’s efforts to discover improved SMOX inhibitors via in silico drug screening and in vitro studies. The experimental details were clearly described and the data supports the author’s conclusions.

I support the publication of this manuscript with minor revisions.

My main critique (and curiosity) is that these molecules are ideally designed to bind metal ions (Cu, Ni, Zn, Fe) as tridentate or tetradentate ligands. This begs the question: what, if any, metal ions are present during the assays used to measure viability and ROS readouts? Metal sequestration or complexation could explain the rapid outcomes seen in Figure 5 (panels A and B), where little to no change in luminol derived luminescence signal is observed even at times near time zero.

This line of critique leads to an alternative interpretation of the data presented, where these compounds bind and sequester a key metal ion needed to perform the assay oxidative step in SMOX assay (or reductive steps in MTS assay) …. rather than inhibiting SMOX directly. This possibility can be excluded by testing common metal binding ligands like Ethylenediaminetetraacetic acid (EDTA), deferoxamine (DFO), catechol, 1,3-propane-diamine, or 2-hydroxybenzyl alcohol as candidate available metal ligands to see how they perform in the SMOX assay. These additional controls (especially the latter two) could be used to rule out/rule metal complexation as a possible alternative explanation.

Relevant refs: Cytotoxicity studies, DNA interaction and protein binding of new Al (III), Ga (III) and In (III) complexes with 5-hydroxyflavone. Munteanu, A-C. et al. Appl Organometal Chem. 2018; e4579. https://doi.org/10.1002/aoc.4579. Note this could explain why 7 is so toxic as it places a OH adjacent to the benzylic alcohol creating a bidentate ligand there whereas 6 does not.

Isatin-Schiff base-copper (II) complex induces cell death in p53-positive tumors. Bulatov, E. Cell Death Discovery volume 4, Article number: 103 (2018)

Metal ion effects on Polyphenol Oxidase Covalently immobilized on a Bio-Composite. Cell Mol Biol (Noisy-le-grand) . 2021 Sep 29;67(2):50-55.  doi: 10.14715/cmb/2021.67.2.8.

Other points to consider: Were any color changes noted upon addition of the compound(s) to the media? Bright red typically indicates an iron complex, for example. Since the MTS reagent is a colorimetric assay, metal complexes could complicate the data obtained if the complex absorbed light near the analytical wavelength used (410 nm). Also is compound 6 structurally modified during SMOX incubation?

Minor edits suggested:

Figure 2 -clipped off box framing at the bottom of the image… is the Figure inside slightly too big?

Page 6: line 15 from top: 0-10 mM inhibitor?? (is this correct, seems high or is this the normal range for the lab for evaluating new compounds)

Note: Typical MTS assay protocols suggest a 4 h incubation after addition of the reagent but three hours is also acceptable as most of the new signal appears by that time point. (see: https://www.abcam.com/mts-assay-kit-cell-proliferation-colorimetric-ab197010.html)

Page 6, line 12 from bottom: …stress, the CM-H2DCFDA

Page 6, line 6 from the bottom: Define the acronym ‘DCF’

Please list the exact 96 well plates used in the plate reader expt (vendor and cat# , black walled?, i.e. were the plates clear, translucent (white) or black (preferred for fluorescence and luminescent expts)?

Page 7: Figure 4. panel B: substituents

Page 9: expand Table 1 margins or spacing or font size so that the numbers do not crunch up and stack. Round up to one or 2 decimal places. Consider not rewriting µM each time as it is already defined in the top line of Table 1 as the IC50 in µM

Page 10: Figure 6: round up IC50 values in the legend of Figure 6 to one or two decimal places not 3.

Page 11: Figure 7: (legend) bold 7 ….compound 7 in lowest….

Page 12: Figure 8: 12 h post treatment seems to be too short a time period. Do the untreated cell double in this time period? If not then the control signal will not change much and if one calculates the change in MTS signal and subtracts time 12h – time zero as a change in growth  then this relative new growth value (in absorbance) will likely be too small (with lots of variability?). For this reason, cytotoxicity screens are typically performed over a period of time where the cells in question double in population. What is the doubling time of SH-SY5Y?

To see a measurable effect in such a short time period, one will need a lot of compound present. This is likely the reason why compounds 4 and 6 are less toxic. Consider performing the toxicity expt over 24h and/or 48h time periods as many compounds require longer time of exposure in order to see a statistically significant effect on cell growth.

Page 12: Figure 9 legend: Why is glutamate listed at 80 mM?? (should this be 8 mM?).  Such a high conc may introduce bias in the system as it creates such a dramatic ‘beatdown’ of viability % and the cells have a hard time recovering. I suggest the authors determine the IC50 of glutamate over 24 hours and then repeat the expt at that concentration. My fear is that the glutamate conc is too high and we are seeing a muted rescue effect, especially with compound 6.  One could then titrate in a higher conc of 6 to determine how much compd 6 is needed to obtain 75% viability (halfway between the glutamate (dosed at its IC50 so 50% rel growth) and untreated controls (100% viability or rel growth) and represent this conc as an EC50 value. The effective conc needed to attain a relative growth halfway between the glutamate and untreated controls. In other words, showing a concentration dependent activity in terms of how compd 6 rescues growth (or viability)  of glutamate treated cells.

I think Figure 7 should also show how compound 6 fits into the SMOX model (cartoon) in panel B as that is the final molecule selected by the authors for future work as readers will be curious to see how it too fits into the SMOX target.

Page 13: Figure 10: add in:  what cell line and over what time period was this performed?

Author Response

Reviewer 1

The manuscript entitled, “Identification and characterization of novel small-molecule SMOX inhibitors” by Woster et al is well written and describes the author’s efforts to discover improved SMOX inhibitors via in silico drug screening and in vitro studies. The experimental details were clearly described and the data supports the author’s conclusions.

I support the publication of this manuscript with minor revisions.

My main critique (and curiosity) is that these molecules are ideally designed to bind metal ions (Cu, Ni, Zn, Fe) as tridentate or tetradentate ligands. This begs the question: what, if any, metal ions are present during the assays used to measure viability and ROS readouts? Metal sequestration or complexation could explain the rapid outcomes seen in Figure 5 (panels A and B), where little to no change in luminol derived luminescence signal is observed even at times near time zero. 

We understand this concern, however we believe metal chelation is not a confounding variable in the luminol assay. Flavin-dependent enzymes such as SMOX and LSD1 do not require metal cofactors for activity, and the assay mixture does not contain any exogenous metals. If chelation was a factor we would have observed the same effect of each compound in the LSD1 and MAO assays. The coupled enzyme horseradish peroxidase does contain an iron molecule, but it is tightly bound in the heme structure and is not likely to be chelated (if chelation was an issue, spermine would be more likely to chelate the iron, but this has never been observed). Finally, the assay calls for an appropriate concentration of the substrate spermine, which would be more likely to chelate any metals that the inhibitors we studied.

Other points to consider: Were any color changes noted upon addition of the compound(s) to the media? Bright red typically indicates an iron complex, for example. Since the MTS reagent is a colorimetric assay, metal complexes could complicate the data obtained if the complex absorbed light near the analytical wavelength used (410 nm). Also is compound 6 structurally modified during SMOX incubation?

We did not observe any color change when compounds were added to the media.

Minor edits suggested:

Figure 2 -clipped off box framing at the bottom of the image… is the Figure inside slightly too big?

Corrected.

Page 6: line 15 from top: 0-10 mM inhibitor?? (is this correct, seems high or is this the normal range for the lab for evaluating new compounds)

Cytotoxicity assays were performed to determine at what concentration compounds themselves exhibit acute cytotoxicity. In this setting, compound concentration is increased until either (a) toxicity is observed or (b) factors of solubility prohibit further dose escalation. In this case, inhibitor concentrations up to 10 mM were required to achieve a significant decrease in cell viability.

Note: Typical MTS assay protocols suggest a 4 h incubation after addition of the reagent but three hours is also acceptable as most of the new signal appears by that time point. (see: https://www.abcam.com/mts-assay-kit-cell-proliferation-colorimetric-ab197010.html)

Readout for MTS assay was monitored at timepoints up to 4 hours following the addition of MTS reagent. When normalized to baseline, there was no significant difference between the 3-hour timepoints and the 4-hour timepoints for this assay. As such, we chose to report the first timepoint at which signal appeared to stabilize. This protocol was used for all MTS assay procedures.

Page 6, line 12 from bottom: …stress, the CM-H2DCFDA

Corrected.

Page 6, line 6 from the bottom: Define the acronym ‘DCF’ 

Done.

Please list the exact 96 well plates used in the plate reader expt (vendor and cat# , black walled?, i.e. were the plates clear, translucent (white) or black (preferred for fluorescence and luminescent expts)?

MTS: (Costar #9018; clear flat bottom); CM-H2DCFDA: (Greiner #655086; black walled, cleat flat bottom); Luminol-HRP: (Costar #3912; white, flat bottom). These descriptions have been added to the text.

Page 7: Figure 4. panel B: substituents

This typo has been corrected.

Page 9: expand Table 1 margins or spacing or font size so that the numbers do not crunch up and stack. Round up to one or 2 decimal places. Consider not rewriting µM each time as it is already defined in the top line of Table 1 as the IC50 in µM

These changes have been made.

Page 10: Figure 6: round up IC50 values in the legend of Figure 6 to one or two decimal places not 3.

This change has been made.

Page 11: Figure 7: (legend) bold 7 ….compound in lowest….

This change has been made.

Page 12: Figure 8: 12 h post treatment seems to be too short a time period. Do the untreated cell double in this time period?

The doubling time of SH-SY5Y cells is reported to be approximately 48 hours. The 12-hour time point was selected as we wanted to observe acute cytotoxicity rather than cellular stasis or changes in cell growth. For this experiment, compounds 4, 6 and 7 (IC50 values 0.23, 0.54 and 0.23 µM, respectively) were dosed at levels that were 40- to 40,000-fold higher than their enzymatic IC50 values. At these high dose levels, compounds 4 and 6 did not produce significant toxicity after 12-hour exposure at doses below 1 mM. This rationale has been added to the text describing Figure 8.

Page 12: Figure 9 legend: Why is glutamate listed at 80 mM?? (should this be 8 mM?).   

The reported 80 mM glutamate is correct. A concentration of 80 mM was selected as this is the reported LC50 of glutamate in SH-SY5Y cells and concentrations in this range are commonly used when investigating cytotoxicity in this cell line. Of note, the LC50 of glutamate is higher in this cell line when compared to other neuronal cell lines, which is thought to be due to the fact that glutamate cytotoxicity in SH-SY5Y cells is mediated primarily through oxidative damage rather than as a direct result of calcium influx. This observation was one of the reasons we selected this particular cell line, given that we are focusing SMOX, which would primarily influence oxidative stress rather than calcium homeostasis.

I think Figure 7 should also show how compound 6 fits into the SMOX model (cartoon) in panel B as that is the final molecule selected by the authors for future work as readers will be curious to see how it too fits into the SMOX target.

We have added representations of the binding of 6 to the figure alongside that of compound 7.

Page 13: Figure 10: add in:  what cell line and over what time period was this performed?

These studies were designed to model acute glutamate-induced toxicity after 1 hour in SH-SY5Y cells. This has been clarified in the text and the legend for Figure 10.

Reviewer 2 Report

This is an elegant paper that shows the identification of inhibitors of SMOX enzyme to be used as chemical probe for further investigations. Although the crystal structure of SMOX is not yet available, the authors overcome this limitation.

1- I have one concern regarding the assertion that compound 6 has a good selectivity profile, as in Figure 6B all the compounds seems to inhibit all the tested enzymes. The authors should clarify this point. 

2- Figure 6A. Are the concentrations reported as absolute numbers or as logarithm? To get a sigmoid curve, the scale of that graph should be in logarithm. Please, clarify. 

3- Figure 6. The letters of the panel figure are missing. 

Author Response

Reviewer 2

This is an elegant paper that shows the identification of inhibitors of SMOX enzyme to be used as chemical probe for further investigations. Although the crystal structure of SMOX is not yet available, the authors overcome this limitation.

1- I have one concern regarding the assertion that compound 6 has a good selectivity profile, as in Figure 6B all the compounds seem to inhibit all the tested enzymes. The authors should clarify this point. 

In drug discovery, determination of selectivity between target enzymes and homologous off-target enzymes is routinely determined. In almost every case, there is some activity against one or more of the off-target enzymes. This can be a criterion for rejection of a compound for consideration. For example, compound 6 is highly selective, while compound 7 is not.

2- Figure 6A. Are the concentrations reported as absolute numbers or as logarithm? To get a sigmoid curve, the scale of that graph should be in logarithm. Please, clarify. 

The X-axis is a log scale.

3- Figure 6. The letters of the panel figure are missing. 

Corrected.
